# Micro-Shear Bands and Their Enhancement on High Temperature Strength of Mg-Gd-Y-Zr Alloy

**DOI:** 10.3390/ma14123262

**Published:** 2021-06-12

**Authors:** Hongchao Xiao, Zhengjiang Yang, Jie Li, Yingchun Wan

**Affiliations:** 1School of Metallurgy and Environment, Central South University, Changsha 410083, China; hcxiao@csu.edu.cn (H.X.); lijieyejin@csu.edu.cn (J.L.); 2School of Materials Science and Engineering, Central South University, Changsha 410083, China; zhengjiangyang@csu.edu.cn; 3Light Alloy Research Institute, Central South University, Changsha 410083, China

**Keywords:** micro-shear bands, strength, Mg-Gd-Y-Zr alloy, dislocation

## Abstract

When Mg-Gd-Y-Zr alloy is cold forged, a large number of nano-micro shear bands are formed inside the grains. It is observed that micro-shear bands hinder the sliding of dislocations, resulting in an increase in tensile strength at elevated temperatures. The subsequent aging treatment further strengthens the alloy. Compared with unforged aged alloys, aged samples with pre-generated micro-shear bands exhibit higher strength at room temperature to 250 °C, but exhibit similar properties at higher temperatures. Microstructure characterization and fracture behavior analysis indicate that the transformation of deformation mode from dislocation sliding to grain boundary activity is mainly due to the change of mechanical properties with temperature. In addition, the alloy precipitates with the aid of dislocations during tension, and exhibits higher strength at 200 °C than that at room temperature.

## 1. Introduction

Mg-Gd-Y-Zr alloys are promising lightweight structural materials in the aerospace industry owing to their super high strength and creep resistance, in addition to the common advantages of Mg alloys such as low density, high specific strength, and ideal recyclability [1,2,3,4]. The excellent performance of Mg-Gd-Y-Zr alloys in terms of strength and creep resistance is attributed to the precipitates formed during the ageing treatment. For example, the plate-like β′ phase particles formed on prismatic planes can provide effective barriers for both dislocation sliding and creep deformation [5,6]. However, aging treatment usually results in a dramatic deterioration in the ductility of the alloy, especially at ambient temperature, leading to the elongation at failure rarely exceeding 6% [7]. Recently, great efforts have been made to overcome the severe trade-off in strength and ductility of the aged Mg-Gd-Y-Zr alloys via developing new enhancement methods. Facts have proven that microstructure modification and texture adjustment by generating ultrafine grain structures can effectively solve this problem. Compared with aging treatment, the formations of ultrafine grains or sub grains in hundreds of nanometer [8], densely distributed nano-separated stacking faults [9], and cell structures [10] are all conducive to obtaining excellent mechanical performance. Our recent study observed the micro-shear bands formed inside the grains and preliminarily confirmed their strengthening effect on the Mg-Gd-Y-Zr alloy [11].

Micro-shear bands are usually formed in body-centered cubic (BCC) and face-centered cubic (FCC) metals with high deformation. Investigation on Ta has indicated that when a dislocation slip system dominates the local deformation, micro-shear bands are formed [12]. Research on Nb has demonstrated that they are formed by dislocations nucleating in pairs, moving in opposite directions, and being accumulated [13]. The micro-shear bands are considered to have a great enhancement effect on the Al-Co-Cr-Fe-Ni high-entropy alloy [14]. Recent investigations have also revealed micro-shear bands observed in Mg alloys processed by severe plastic deformation [5,15]. However, there are few reports on their effect on the mechanical properties of Mg alloys.

In this work, we observed the formation of micro-shear bands in cold forged Mg-Gd-Y-Zr alloy. The mechanical properties of the alloy at elevated temperatures with and without micro-shear bands and before and after aging treatment were systematically investigated. The results can provide guidance for the design of microstructures with preferable mechanical properties in other Mg alloys.

## 2. Experimental Procedures

The Mg-8Gd-3Y-0.4Zr (wt.%) alloy obtained by semi-continuous casting, homogenization, and hot extrusion was used. The microstructure of the as-used alloy has been revealed in our previous study. The alloy was then cold forged to a total deformation of 7% and 12%, and then some of them were aged. The samples are respectively named according to their processing and treatment experiences: AE for extruded alloys, F-7 for forged alloys with a deformation of 7%, F-12 for alloys with a deformation of 12%, AE-A for aged AE, and F-12-A for aged F-12. AE-A and F-12-A were both aged at 200 °C for 40 h. The microstructure characterization and mechanical testing were carried out on the samples obtained in the central regions of the processed alloy.

Transmission electron microscopy (TEM) characterizations were performed using a JEOL F200 microscope (JEOL, Tokyo, Japan) operating at 200 kV. The samples for TEM and HRTEM characterization were prepared by twinjet electropolishing in a solution of 97% alcohol and 3% nitride acid at −35 °C.

Tensile tests were conducted on an Instron 3369 electronic universal machine (Instron, Boston, MA, USA) at room temperature and elevated temperatures at a strain rate of 10^−3^·s^−1^. According to the ASTM procedure, the specimens used for tensile tests were machined into a gauge radius of 5 mm and a gauge length of 25 mm. All mechanical properties were taken from the arithmetic mean of three parallel samples. The hardness tests were conducted on a Vickers hardness tester (Yanrun, Shanghai, China) (with a load of 4.9 N and a dwelling time of 15 s. In order to ensure reliability, at least 5 indentations were made for each test, and the average values were recorded. After tension, the scanning electron microscope observation of the fracture surface was conducted on an FEI Helios NanoLab 600i microscope (FEI, USA).

## 3. Results

### 3.1. Microstructure

The TEM bright field (BF) image depicted in Figure 1 shows that considerable micro-shear bands were generated after cold forging with thicknesses ranging from tens of nanometers to hundreds of nanometers and lengths in the range of a few micrometers. The number of micro-shear bands obtained during the forging process is related to the strain level we previously studied [11]. The selected area electron diffraction (SAED) in the inset reveals that the misorientation between the banded region and the matrix is between 3 and 15 degrees. It is worth noting that these bands are frequently observed near the initial grain boundaries with a strong contrast of dislocations observed around them.

From our previous study, it can be seen that the micro-shear bands have an obvious strengthening effect on the alloy at room temperature [11]. Since the Mg-Gd-Y-Zr alloys were designed for use at elevated temperatures, it is necessary to study how the micro-shear bands affect the high-temperature mechanical properties of the alloy.

### 3.2. Mechanical Properties

In order to determine the influence of cold forging, especially the formation of micro-shear bands, on the mechanical properties of the alloy at elevated temperatures, tensile tests at 200 °C were carried out on the alloy under various conditions with the results shown in Table 1. The extruded alloy had a relatively low yield strength of about 197 MPa and an elongation at failure of 29%. Cold forging led to a significant increase in strength and a significant decrease in ductility: the yield strength of F-7 samples and F-12 samples increased by 85 MPa (43%) and 140 MPa (71%), respectively, and the elongation at failure decreased to about half that of the sample AE.

Mg-Gd-Y-Zr alloy is a typical age-hardening alloy which is achieved by forming precipitates on the prismatic planes. The precipitation behavior of the alloy can be tailored by introducing defects in advance or altering the aging process, and its mechanical performance can be greatly tuned. The pre-formed dislocation arrays will help to generate bamboo-like precipitation bands with precipitation-free zones, as the dislocations are the preferred precipitation sites for the β′ phase [16]. Thus, we further investigated the effect of the introduction of micro-shear bands on the aging response and mechanical properties of the alloy.

As shown in Figure 2, when aging at 200 °C, the alloy exhibits a clear hardening response and reaches peak hardness after 40 h. 

Compared with the fully dynamically recrystallized counterpart of this alloy which reached peak hardness only after 64 h at the same temperature [17], a higher hardening response was obtained, revealing an accelerated precipitation kinetic in the forged alloy. However, the hardening response at peak hardness (20HV) is lower than that in the extruded alloy (45HV) [18]. As the alloy has characteristics of randomly distributed dislocations and micro-shear bands, further investigation was carried out by TEM to uncover whether the micro-shear bands have an additional acceleration effect on the precipitation of the alloy at 200 °C.

Figure 3 shows the TEM BF microstructure of the alloys under forging conditions and after aging at 200 °C for 40 h, with the SAED pattern in the upper left corner of the insert. Only the Mg pattern was observed in the forged alloy, and the planes are indicated by the white text in Figure 3a. After aging, the precipitates were observed to be almost uniformly distributed in the alloy, as shown in Figure 3b. The SAED pattern has three additional dots as shown in orange, indicating that the precipitates are the β′ phase [19]. No obviously larger size or denser distribution were observed at the band boundaries. This result demonstrates that the band boundaries of the alloy at 200 °C have no further precipitation enhancement. The advanced hardening response is caused by all the dislocations inside and outside the bands and at the boundaries.

We conducted tensile tests in the temperature range of 20 °C to 300 °C to further study the effect of aging treatment (peak-hardness treatment) after cold forging on the mechanical properties of the alloy. As shown in Table 2 and Figure 4, the results show that aging treatment enhances the extruded and forged alloys.

Compared with previously reported samples without aging treatment [11], at room temperature the enhancements for the extruded alloy and forged alloy are 81 MPa (301–220) and 51 MPa (386–335), respectively, and at 200 °C, the enhancements are 118 MPa (315–197) and 65 MPa (402–337), respectively. It is worth noting that the aging enhancement of the forged alloy is much lower than that of the extruded alloy, which is consistent with the hardening response. This may be attributed to two reasons: (1) the recovery has softened the alloy; (2) the precipitation behavior was tailored according to the resultant undesirable size or distribution of the precipitates. Thus, it can be inferred that at 20–250 °C, the strength of sample F-12-A is higher than that of AE-A, which is also caused by the micro-shear bands.

More interestingly, it was observed that the yield strength of the alloy first increased and then decreased with temperature, and it had the highest yield strength at 200 °C. In other Mg-Gd-Y-Zr alloys, this kind of abnormal strength at elevated temperature higher than that at room temperature has also been observed [18,20]. In addition, at a temperature of 20–250 °C, the strength of F-12-A alloy was obviously higher than that of AE-A alloy, but it had close performance at 300 °C. In fact, the strength difference between the two samples started to decrease at 250 °C (85 MPa at room temperature, 87 MPa at 200 °C, and 69 MPa at 250 °C). These two features are considered to be the result of changes in dominant deformation mode at different temperatures.

## 4. Discussion

In order to reveal the strengthening mechanism at elevated temperature, the microstructure of sample F-12 after being tensioned to 7% strain at 200 °C was characterized and is shown in Figure 5.

Compared with the micro-shear bands in the un-tensioned alloy shown in Figure 1 and Figure 3a, the bands here have obvious curving features, as indicated by the red dashed lines. It is observed that the number of dislocations end at the band boundaries (indicated by the red arrows), indicating the boundaries effectively hinder the movement of the dislocations. This barrier mechanism is similar to that provided by grain boundaries, and logically follows the Hall-Petch type relationship. The trouble here is that it is difficult to evaluate the mean boundary distance of the micro bands, as they have a rather uneven distribution within a single grain and between different grains. Testing on microsamples containing several initial grains with a size of tens of micrometers will help to further determine the quantitative relationship between strength and boundary distance. Even so, we can conclude that the effective strengthening effect of cold forging on Mg-Gd-Y-Zr alloy at 200 °C is largely due to the formation of micro-shear boundaries, which act by preventing dislocation sliding.

Mg-Gd-Y-Zr alloys at 200 °C usually exhibit higher yield strength and ductility than that at room temperature under various conditions, such as DRXed, aging, and with micro-shear bands [21,22]. In order to reveal the underlying mechanism, we further investigated the deformation mechanism of the alloy at 200 °C. To minimize the interference from the pre-existing precipitates and bands, the extruded alloy with DRXed microstructure and no precipitates was used. The TEM BF images of the tensioned sample is depicted in Figure 6.

Figure 6a, using the g vector of [0001] (shown in the upper left corner), reveals the number of dislocations in the sample. Their visibility in this condition demonstrates that their Burgers vectors have a <c> component, and most of them or their projections are parallel to various non-basal planes. These sliding characteristics indicate that the dominant deformation mechanism of the alloy at 200 °C is similar to that at room temperature. However, the dislocations or projections in Figure 6a are much shorter than in Figure 6a obtained under the same diffraction conditions. The very fragmented features of dislocations may be due to cross-slipping between pyramidal planes or being hindered by particles. Then, we further characterized the sample from a full diffraction pattern and observed precipitates as marked by the blue arrows in Figure 6b. The SAED pattern on the upper left corner indicates that they are β″ phase, as shown in orange text [23]. When the temperature of Mg-Gd-Y-Zr alloys is maintained in a range of 175–350 °C, precipitation occurs, and its solid solution concentration is higher than the solubility limit. The order of precipitation is (S.S.S.S.) → β″ (DO19) → β′ (CBCO) → β_1_ (FCC) → β (FCC), and β″ phase is observed in the very initial stage of the aging process (after 0.5 h at 225 °C) [24]. However, alloy precipitation can be greatly enhanced by pre-induced defects or stress applied along aging [25]. In this tensioned sample, the stress is considered to accelerate the precipitation kinetic and induce the rapid formation of β″ phase. Then, the obtained β″ phase particles hinder the dislocations sliding, resulting in the fragmented feature of the dislocations as shown in Figure 6a. The resistance of the newly formed phase particles to slipping resulted in higher flow stress, thus maintaining the deformation. Therefore, the alloy has higher strength at 200 °C than that at room temperature.

In order to further investigate the deformation mechanism and understand the mechanical properties of the alloy when tensioned at elevated temperature, we characterized the fracture surfaces of the tensioned samples via SEM. Figure 7 presents the fracture characteristics of the wrought samples tensioned at 200 °C.

In this extruded alloy, nearly all the fractured surfaces are covered with dimples and tear ridges. While in the as-forged alloy, apart from dimples, cleavage planes are also observed. These characteristics indicate that ductile fracture is dominant in the failure behavior of the extruded alloy, while mixed fracture occurred in the forged alloy. The extruded alloy shows the deepest dimples, while the forged alloy exhibits shallower ones, indicating that a larger uniform post-elongation occurred in the extruded alloy, causing the dimples to become slenderer. Compared with RF-7, the fracture surface of sample RF-12 has a larger area covered by the cleavage planes. These characteristics demonstrate that the ductility of the alloy decreases with pre-forging strain.

Figure 8 shows the fracture surfaces of the F-12-A samples tensioned at 200 °C, 250 °C, and 300 °C.

Dimples, cleavage planes, and tear ridges are observed on the fracture surface of the sample tensioned at 200 °C, revealing that mix-mode fracture dominates the failure behavior. As the temperature increases, the fraction of dimples increases, indicating that the ductility is improved. Similar to Figure 7, the size of dimples formed at 200 °C varies in the range of 2–10 μm. When temperature rises to and exceeds 250 °C, the dimples obtained are obviously larger. The size of the dimples generated at 300 °C is equivalent to the grains, and steps are observed on the dimple walls, indicating that the grain boundary activity was activated. The fracture surface characteristics of the alloys demonstrate that the plastic deformation was accommodated inside the grains at room temperature to 200 °C and was dominated by the grain boundary activity when the temperature was increased to 300 °C. As micro-shear bands were observed to effectively hinder dislocations sliding, thus higher strength is obtained when deformation is accommodated inside the grains. In the case of deformation accommodated by the grain boundary, the enhancing factors that play a role in the deformation inside the grains no longer contribute to the strength. Thus, samples with different micro-shear bands exhibit similar strengths at 300 °C. In addition, with the increase of temperature, the mobility of dislocations at the micro-shear boundary increase, thereby producing decreasing effect on hindering dislocation sliding, which also helps to reduce the enhancement at 250 °C.

## 5. Conclusions

In this study, we observed micro-shear bands formed inside the grains of cold-forged Mg-Gd-Y-Zr alloy and then investigated their effects on the high-temperature mechanical performances of the alloy. It provides a new way to modify the mechanical property of alloys by introducing micro-shear bands. The following conclusions can be reached:The micro-shear bands can effectively hinder dislocation sliding and strengthen the alloy at elevated temperatures. When the temperature is higher than 250 °C, the enhancement effect decreases, which is mainly caused by the activation of grain boundary deformation.Due to dislocation-assisted precipitation during tension, the alloy exhibits higher strength at 200 °C than at room temperature.

## Figures and Tables

**Figure 1 materials-14-03262-f001:**
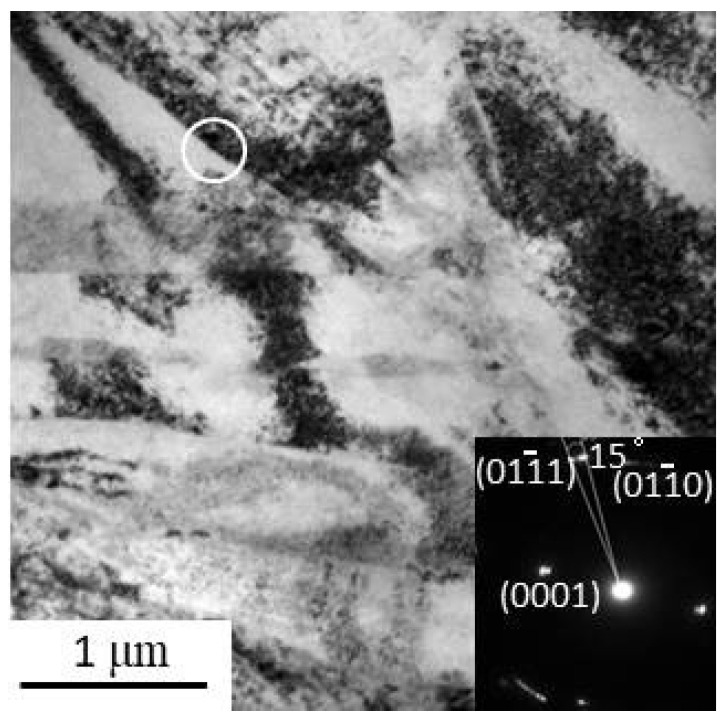
Transmission electron microscopy bright field microstructure and selected area electron diffraction pattern of sample F-12.

**Figure 2 materials-14-03262-f002:**
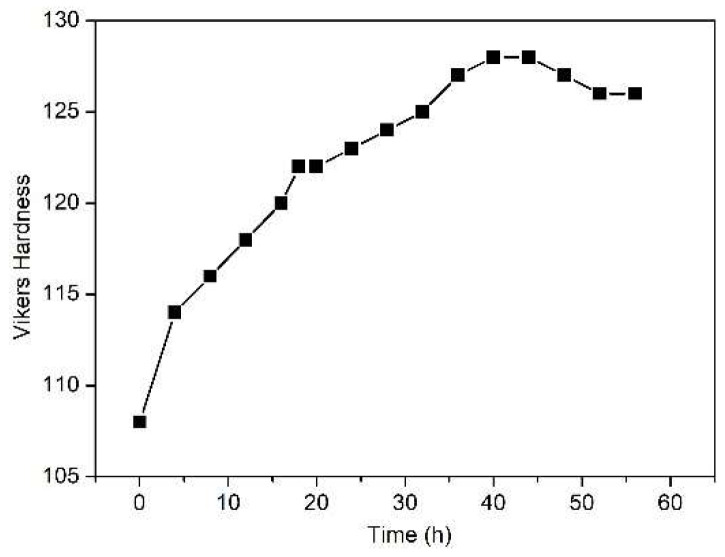
Age-hardening response of sample F-12.

**Figure 3 materials-14-03262-f003:**
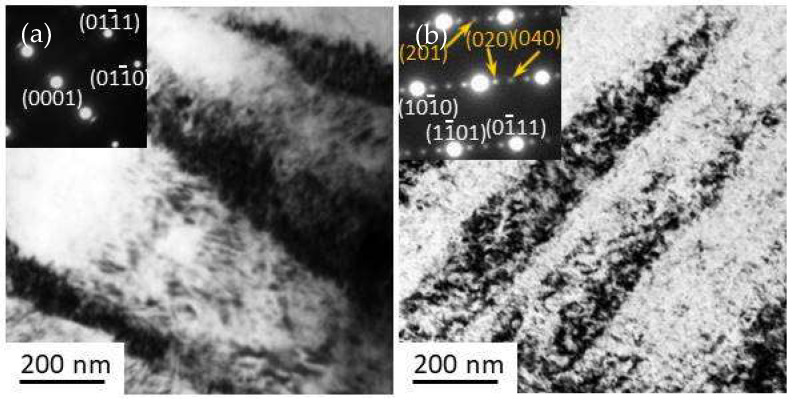
Transmission electron microscopy microstructure of sample F-12-A. (**a**) the forged alloy, (**b**) the aged alloy.

**Figure 4 materials-14-03262-f004:**
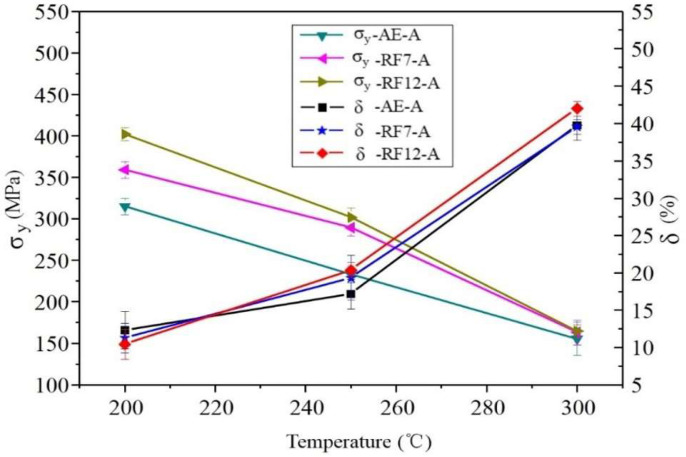
Yield strength and elongation at failure varying with temperature.

**Figure 5 materials-14-03262-f005:**
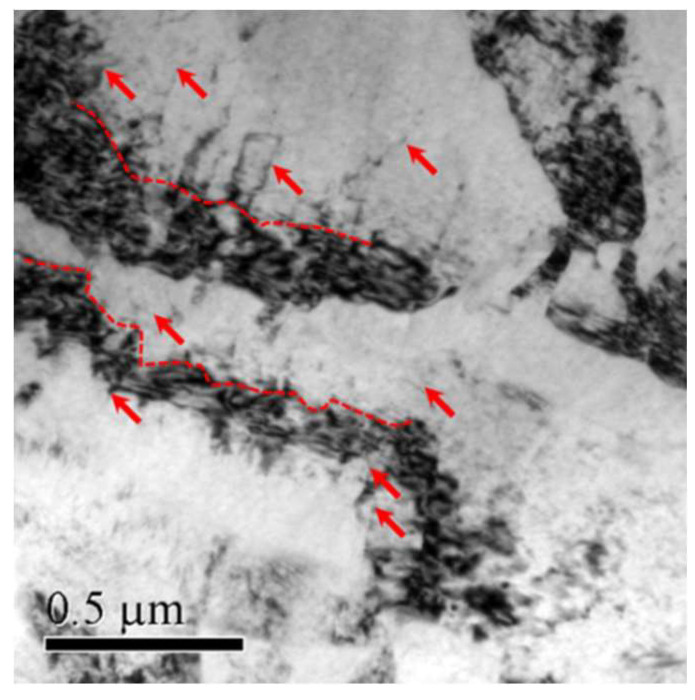
Transmission electron microscopy image of sample F-12 tensioned at 200 °C.

**Figure 6 materials-14-03262-f006:**
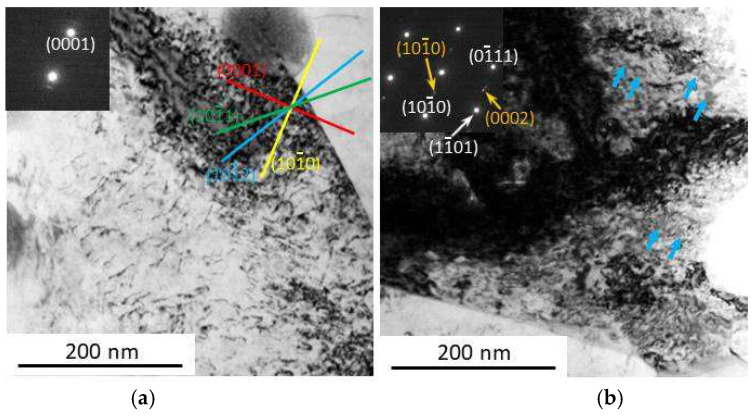
Transmission electron microscopy image of sample AE tensioned at 200 °C. (**a**) with g vector of [0001] (**b**) the SAED pattern obtained when the electron beam is parallel with [1$\bar{2}13$].

**Figure 7 materials-14-03262-f007:**
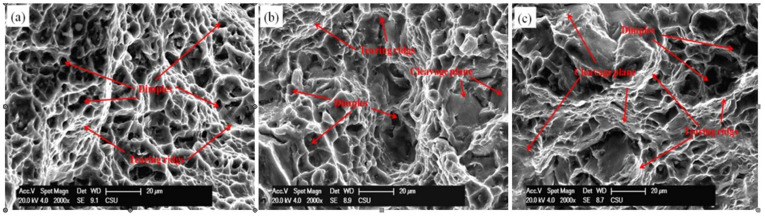
Scanning electron microscopy images of fracture surfaces of different samples at 200 °C (**a**) AE, (**b**) F-7, (**c**) F-12.

**Figure 8 materials-14-03262-f008:**
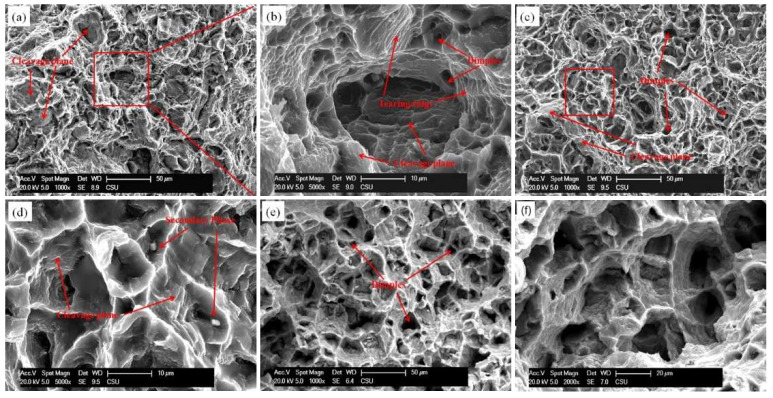
Scanning electron microscopy images of the fracture surfaces of sample F-12-A at different temperatures: (**a**,**b**) 200 °C, (**c**,**d**) 250 °C, and (**e**,**f**) 300 °C.

**Table 1 materials-14-03262-t001:** Tensile mechanical properties at 200 °C.

Samples	σ_y_ (MPa)	σ_u_ (MPa)	Δ (%)
AE	197	282	29
F-7	282	332	15
F-12	337	360	14

**Table 2 materials-14-03262-t002:** Tensile mechanical properties of the aged alloys.

Samples	Temperature (°C)	σ_y_ (MPa)	σ_u_ (MPa)	Δ (%)
AE-A	20	301	430	4.7 [10]
	200	315	377	12.0
	250	233	299	17.0
	300	156	185	40.0
F-12-A	20	386	500	4.0
	200	402	450	10.4
	250	302	334	20.4
	300	165	198	42.0

## Data Availability

Data available in a publicly accessible repository that does not issue DOIs. Publicly available datasets were analyzed in this study.

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
