# Peer review of "Micro-Shear Bands and Their Enhancement on High Temperature Strength of Mg-Gd-Y-Zr Alloy"

_materials, 2021, doi:10.3390/ma14123262_

Round 1

Reviewer 1 Report

The paper touches on a significant problem with promising lightweight Mg-Gd-Y-Zr alloy. Nano and micro shear bands occurring in this alloy play an important part in in the material properties (like tensile strength) in high temperatures. In my opinion this scientific paper should be considered for publication in the Journal.

Saying that, there are few things that need to be changed before the publication of this paper:

  • Table 1 and 2 are to (they should be as width as the text)
  • Same with figures 7 and 8
  • In the conclusions, it would be good to add one that will recommend something to the end user or producer of the alloy based on the information gathered during the research for this paper.
  • The Mono-ethnicity of the bibliography (although somewhat justified) is a little bit unsettling. Are there really no scientist  form Eurepe, North or South America that worked on Nano and micro shear bands, Mg-Gd-Y-Zr alloy or high temperature material properties changes? Maybe some of them did that before authors of the cited papers? Please consider extending the bibliography.
  • Also is the bibliography written in official MDPI format?

Regards,

 Author Response

  • Table 1 and 2 are to (they should be as width as the text)

They have been modified.

  • Same with figures 7 and 8.

They have been modified.

  • In the conclusions, it would be good to add one that will recommend something to the end user or producer of the alloy based on the information gathered during the research for this paper.

It has been modified. We added a sentence “It provides a new way in modifying the mechanical property of alloys by introducing micro-shear bands.”

  • Also is the bibliography written in official MDPI format?

It has been modified.

Reviewer 2 Report

The article contains some new results, but cannot be accepted for publication in present form and requires substantial revision, including English quality (see below).

Detailed questions and comments:

Introduction

- It is not clear why all sections of the article have the first number?

- Since, as the authors note, they studied the formation of micro-shear bands and their effect on strength earlier (ref. [11]), it is necessary to clarify the purpose of this work. What is interesting in the mechanical properties of the alloy at elevated temperatures?

Experimental

- Which regimes and what equipment were used for forging of the magnesium alloy?

- Why such low strain degrees (7 and 12%) were chosen? What size of billets was used?

- Why does the text contain information on the HRTEM mode, the results of which are not presented in the work?

- How were samples cut and prepared for mechanical testing?

- What equipment was used for hardness measurements?

- From what area the SAED patterns were obtained by TEM studies?

Results

- It is necessary to provide data about the initial structure of the Mg alloy (grain size, dislocation density) and how it was changed after forging.

- To analyze the mechanical behavior of the alloy it is necessary to provide tensile curves (!)

- What is the chemical composition of the β'-phase? Bring it (!)

- On page 4, fig. 4 wrongly mentioned instead of fig. 2 (line 114).

- In the caption to fig. 2 does not indicate the temperature at which aging was carried out.

- Why in fig. 4 values ​​of yield strength and elongation to fracture are given for different test temperatures (?!) How these values were ​​obtained (?!)

- Why has the effect of aging at 200 ° C on hardness been studied only for F12 (?!) It is also necessary to provide data for the initial state (AE) and forging 7% (F7)!

- In the text of the article there are no refs to Fig. 2 and 3 (?!) In general, most of the links in the text to figures are not correct (!)

- In the caption to Figure 3, there is no information about the temperature and aging time, as well as about the difference between figures a and b (?!)

- In the pic. 2 particles of the β'-phase are not indicated (?!)

- The caption to Table 2 does not indicate what was the duration of aging (?!)

- The expression on page 5, lines 149-150 is not entirely clear and requires revision.

- The text of the article (lines 163-165) does not correspond to Figure 7 (!)

- The sentence on page 6, (lines 180-182) is not clear and needs reformulation.

- What is the component of the Burgers vector (?)

- What is the chemical composition of the phases indicated on page 6 (β', β'', β1 & β)? What dispersion and volume fraction do they have?

- As noted in the article (P.6, lines 196-198) at temperatures of 175-350 ° C, the solid solution decomposition occurs. However, which elements and phases are precipitated is not indicated (?!)

- What does it mean "fragmented feature of the dislocation" (P. 6, lines 203-205)?

- If, as the authors believe, an increase in the strength properties of the alloy is associated with the precipitation of β'' particles during deformation at T = 200 ° C (which are not indicated in Fig. 6b) than what an increase in strength properties of the alloy after aging is caused, where decomposition has already occurred with precipitation of β phase particles?

- Legends in fig. 7 and 8, rendered in red color, are very hard to see (?!)

- What does the term “uniform post-elongation” (P.7, line 224) mean (?!)

- What are the designations RF-7 and RF-12 given on P.7, line 226 (?!)

- What is the basis for the authors' assertion that the steps on the dimples indicate the development of “grain-boundary activity” (?!)

- In order to confirm the assertion made by the authors that at temperatures above 200 ° C the deformation mechanism changes from intragranular to grain boundary, it is necessary to bring the fracture surfaces of the samples after tension at a lower, for example, room temperature! Moreover, the development of “grain-boundary activity” observed at T = 200 ° C does not lead to a decrease in the strength properties of the alloy (?!)

Author Response

- Which regimes and what equipment were used for forging of the magnesium alloy?

Rotary swaging was used for the alloy.

- Why such low strain degrees (7 and 12%) were chosen? What size of billets was used?

Micro-shear bands will disappear and nanograins will be formed at higher strain, as has been studied in our previous study “Acta Materialia 200 (2020) 274–286”. The used billets used has a diameter of 18 mm and a length of 1 m.

- Why does the text contain information on the HRTEM mode, the results of which are not presented in the work?

No results were based on HRTEM and it has been deleted.

- How were samples cut and prepared for mechanical testing?

The samples were prepared by mechanical machining.

- What equipment was used for hardness measurements?

The hardness was determined by Vikers micro hardness test.

- From what area the SAED patterns were obtained by TEM studies?

The SAED pattern was obtained in a area with a diameter of 200 nm.

- Why in fig. 4 values ​​of yield strength and elongation to fracture are given for different test temperatures (?!) How these values were ​​obtained (?!)

It has been modified.

- Why has the effect of aging at 200 ° C on hardness been studied only for F12 (?!) It is also necessary to provide data for the initial state (AE) and forging 7% (F7)!

In the present study was focused on the micro-shear bands, sample F12, which has the most micro-shear bands was selected for further study.

- In the text of the article there are no refs to Fig. 2 and 3 (?!) In general, most of the links in the text to figures are not correct (!)

They have been modified.

- In the caption to Figure 3, there is no information about the temperature and aging time, as well as about the difference between figures a and b (?!)

Figure a and b are both for the sample aged at 200℃ for 40 h.

- In the pic. 2 particles of the β'-phase are not indicated (?!)

No β'-phase in Figure 2.

- The caption to Table 2 does not indicate what was the duration of aging (?!)

AE-A and F-12-A are both aged at 200℃ for 40 h, which has been added in Line 62.

- The expression on page 5, lines 149-150 is not entirely clear and requires revision.

- The text of the article (lines 163-165) does not correspond to Figure 7 (!)

It has been modified. Lines 149-150 correspond to Figure 5.

Reviewer 3 Report

The presented work covers a very interesting topic of the influence of micro shear bands on the Mg alloy behavior on elevated temperature. 

However, the work suffers from several drawbacks which need to be improved. 

There many mistype in the text, the presentation of the images is rather poor quality and unclear. 

There is a lack of HRTEM  investigations which are described in the methodological part. 

The introduction part is based on the author's publication which has not been published yet ref.11. 

I suggest improving the publication with those recommendations.

Author Response

There is a lack of HRTEM investigations which are described in the methodological part. 

No results were based on HRTEM investigation and it has been modified.

The introduction part is based on the author's publication which has not been published yet ref.11. 

It has been published now and the information about Ref[11] has been added.

Reviewer 4 Report

Dear authors,

I congratulate you for such interesting work. The results can provide guidance for developing new Mg based alloys with tailored mechanical properties.

I suggest you to give a complete bibliography references on [22] and [24], year of publication is missing.

Also I suggest you to develop more the conclusion part, if possible.

Kind regards

Author Response

I suggest you to give a complete bibliography references on [22] and [24], year of publication is missing.

They have been modified.

Round 2

Reviewer 2 Report

The manuscript was improved and can be accepted for publishing.

Reviewer 3 Report

The paper still suffers from some editing issues. 

Please check the publication once again in order to remove mistakes eg. :

line 106  "precipitation bahavior"  should be behavior

line 135  "ally" should be alloy

wrong reference style and change in font style and size etc.